# Urban Food Systems: A Bibliometric Review from 1991 to 2020

**DOI:** 10.3390/foods10030662

**Published:** 2021-03-19

**Authors:** Qiumeng Zhong, Lan Wang, Shenghui Cui

**Affiliations:** 1Key Lab of Urban Environment and Health, Institute of Urban Environment, Chinese Academy of Sciences, Xiamen 361021, China; qmzhong@iue.ac.cn (Q.Z.); lwang@iue.ac.cn (L.W.); 2University of Chinese Academy of Sciences, Beijing 100049, China; 3Xiamen Key Lab of Urban Metabolism, Xiamen 361021, China

**Keywords:** sustainable food system, network analysis, urban agriculture, sustainability, resilience, food security

## Abstract

The increase of urbanization is affecting the urban food system (UFS) in many areas, primarily production, processing, and consumption. The upgrading of the urban food consumption structure not only puts forward higher food production requirements, but also poses a challenge to resource consumption and technological innovation. Considerable case or review studies have been conducted on UFS, but there is no bibliometric review attempting to provide an objective and comprehensive analysis of the existing articles. In this study, we selected 5360 research publications from the core Web of Science collection from 1991 to 2020, analyzing contributions of countries, institutions, and journals. In addition, based on keyword co-occurrence and clustering analyses, we evaluated the research hotspots of UFS. The results show that global research interest in UFS has increased significantly during these three decades. The USA, China, and the UK are the countries with the highest output and closest collaborations. UFS research involves multiple subject categories, with environmental disciplines becoming mainstream. Food security, food consumption, and food waste are the three main research areas. We suggest that food sustainability and resilience, food innovation, and comparative studies between cities should be given more attention in the future.

## 1. Introduction

As a result of growing population, rapid urbanization, and rising incomes, global food demand is increasing and diets are being upgraded [1]. While global food production has generally kept pace with population growth, approximately 800 million people still lack sufficient food, and 690 million people are undernourished, while at the same time there are high levels of adult obesity (13.1% in 2016) and a rapid increase in childhood obesity as well (5.6% for children under five years old) [2]. The food system is a coupling system that integrates people with the environment, society, and the economy, involving all elements and activities related to food production, processing, transportation, consumption, and waste disposal [3,4]. Food systems are closely linked to several pressing issues such as climate change and resource scarcity. For example, global food production may contribute up to 25% of the total greenhouse gas emissions [5]; the expansion of agricultural land has led to biodiversity loss [6]; about 92% of water consumption is related to agriculture [7]; there is heavy use of fertilizers and pesticides, leading to water pollution [8,9]. Besides, there is massive energy consumption during food processing and transportation [10], and food waste throughout the food system [11]. The food system is subject to conflicting pressures. Increasing food demand, insufficient food supply, extreme weather events and resource destruction threaten the stability of the food system. A range of issues related to food systems has received widespread attention from researchers, the public and governmental organizations. Food systems are interwoven across several United Nations Sustainable Development Goals (UN SDG), notably SDG 2 (zero hunger), SDG 3 (good health and well-being), and SDG 12 (sustainable consumption and production).

Rapid global urbanization has transformed cities into the major population centers, where the majority of people live, produce, and consume. The new urban agenda reaffirms the global commitment to sustainable urban development, recognizing cities as key areas for achieving sustainable development goals [12]; SDG 11 (Sustainable cities and communities) calls for sustainable cities. The acceleration of urbanization has changed the food system in many ways [13], and the most discussed issues in the urban food system are food supply and dietary changes. Urban expansion occurs in some of the most productive farmlands and over large areas [14], posing a significant threat to food production. Urban agriculture is defined as production in the home or in plots within urban or periurban areas, which could increase urban residents’ income and guarantee local food security, and has the environmental advantages of reducing food transportation distances and thereby emissions [15,16,17]. In terms of urban food consumption, with the improvement of living standards, the dietary structure has changed toward higher consumption of oils, meat, and refined sugar [18]. Many researchers have identified this shift and explored its impact from different perspectives, such as health problems directly resulting from dietary changes [19,20] or indirect effects on the environment [21,22,23]. Food waste is also becoming more severe in cities, often during food transportation and retail consumption. Furthermore, the digital revolution has given rise to a new consumption pattern—takeout food. While takeout increases convenience, it generates a large amount of waste—not only plastic packaging and disposable plastic tableware but also leftover food residues that increase the pressure on urban waste disposal systems and cause environmental problems [24,25,26]—that deserve much more attention.

As the number of publications focusing on the urban food system (UFS) has increased dramatically, many review papers have been written on this subject, from a number of research directions, such as the impact of urbanization on food systems [26], policy implications of urban waste [27], and urban food security [28]. However, as the research of UFS involves many disciplines, it is hard to cover the entire spectrum of research areas or to identify the UFS research trends, in a single review article, and there is a large degree of subjectivity in the literature selection [29]. To fill this gap, this study applies bibliometric analysis to provide a more comprehensive and accurate analysis of UFS. Bibliometrics was first proposed by Pritchard in 1969 [30]; it uses mathematical and statistical methods to determine the status and future direction of specific research fields.

The goals of this study include (1) exploring the trends and characteristics of UFS research from 1991 to 2020; (2) analyzing the contributions and collaborations of different countries, institutions and journals to this field, highlighting the most influential papers; (3) revealing the research focus through keyword analysis; (4) identifying the challenges for UFS research.

## 2. Materials and Methods

A systematic bibliometric literature review follows a series of steps. First, topically relevant keywords are needed for searching electronic databases; the Web of Science (WoS) is used for this study. Second, some literature analysis tools (such as Hiscite) are applied for detecting emerging trends in the selected papers. The third step is the application of network analysis tools (such as VOSviewer) to assess the collaborative relationships between countries/regions and institutions.

VOSviewer and Hiscite were selected to conduct the bibliometric analysis for this study [31]. VOSviewer is a software tool for creating bibliometric networks of countries, institutions, journals, researchers, or publications; the networks are based on coauthorship, co-occurrence, or citation. For this study, the tool is particularly useful for exploring cooperation between countries and institutions based on co-occurrence analysis and identifying key research topics through cluster analysis. Hiscite is a citation-based analysis software that can quickly locate important publications in the research direction of interest. In addition, we used several recognized indicators to reflect the papers’ levels of impact. Impact factor (IF) was first mentioned in 1955 [32], and proposed by Eugene Garfield in 1972 [33]. It can describe the impact of a journal, referring to the frequency with which articles of a particular journal are cited in a specific year or period [34]. H-index is a mixed quantitative indicator that can evaluate the amount and level of academic output of researchers. An index of h means that there are h papers that have each been cited at least h times.

WoS was developed by Thomson Scientific, and it covers most scientific fields. It has a quick search, an advanced search, a general search, and a cited reference search, which it uses to compile the literature data set [35]. In this study, two types of keywords used for data collection were “food system$” or (food near/3 (produc* or proces* or transpor* or consump* or wast* or dispos*)) and “city or cites or urban* or downtown or metropolis.” Since there were very few studies on UFS before 1991, the publication years of 1991 to 2020 were chosen (the publication search was conducted in January 2021). Based on the retrieval method mentioned above, a total of 5674 publications were found to be in accordance with the selection criteria. The research results include all the essential publication information (including titles, keywords, abstracts, affiliations, authors, and references). Among these publications, according to categories within the WoS, periodical articles accounted for 78.8%, followed by proceedings papers (12.1%), and reviews (7.3%) (Figure 1). Since articles accounted for the largest share of the total document types, this study only contains articles, proceeding papers, and review papers applicable to our research goals. English (96.3%) is the most frequently used language, followed by Portuguese (1.3%). In this study, only publications in English are selected because of their academic popularity. The final database is composed of 5360 research publications.

## 3. Results

### 3.1. General Performance of Selected Publications

The number and trend of publications generally reflect the extent of attention to a field. Figure 2 presents the total number of publications (TP) on UFS each year from 1991 to 2020, and the study period can be classified into three subperiods: initial period (1991–2000), stable-growth period (2001–2010), and rapid-growth period (2011–2020). In the initial period, the number of publications was relatively small and grew slowly, totaling 268. In the second period, UFS began to receive widespread attention from scholars, showing a clear upward trend. Compared with the initial period, the second period increased by 255.6%, to 953. In the third period, particularly after 2014, the volume of publications increased significantly. The average annual volume of publications of the third period reached 414, and the total number was 4139, an increase of 334.3% compared with the second period. The volume of publications on UFS shows a trend of steady increase every year.

### 3.2. Country/Region Contribution Analysis

A total of 169 countries (or regions) have been represented in UFS research publications from 1991 to 2020. Among these countries, the USA, China, the UK, Brazil, and Australia are the top five most productive countries, accounting for 24.8%, 13.6%, 8.2%, 6.6%, and 6.0% of the total selected publications, respectively. From the perspective of the h-index, the USA (97), the UK (56), China (55), Canada (50), and Australia (46) are the top five countries in terms of influential publications. Table 1 shows the top 20 countries (in number of publications) that participated in ongoing UFS research during the three periods. The composition of the top productive countries shows that developing countries do not publish as much as developed countries do. As for the different periods, the USA has always been in a leading position with the largest number of publications, accounting for 34.3%, 25.9% and 25.8% of the total number of publications, for the first, second and third periods, respectively. As UFS research is limited to urban areas, most countries that publish more UFS articles have higher urbanization levels than the global average. In the initial period, of the top 20 countries, only China, India, Kenya, and Bangladesh were below the world’s average level of urbanization; in the second period, China, India, and Nigeria were below the global level; in the third period, only India was below the global level. In terms of research and development expenditures (% of GDP), the top 20 countries have been increasing their investment in scientific research during 1991–2020, with six countries investing more than 2% in research in the first period, seven countries in the second period, and an increase to 10 countries in the third period. This observation indicates that both the level of urbanization and the research and development expenditures may affect UFS research. However, although the urban food systems may be more fragile in some low-income countries, there are not enough theoretical or practical UFS research results to make research investment worthwhile. China is the fastest-growing country in UFS research; its volume has grown from only four in the initial period to 610 in the rapid-growth period.

Some of these papers are cooperative endeavors across countries. Co-occurrence analysis of the top 50 countries/regions (number of publications) is used for evaluating the collaboration between countries (only articles published in collaboration are included). In Figure 3, the circle size represents the number of partner countries; the color value represents the average number of citations per publication; the thickness of the line segment represents the degree of direct cooperation between two countries/regions. The USA, Germany, the UK, Canada, and China have participated in more international collaborations. Each of these countries/regions has cooperated with 49, 49, 47, 46, and 45 of the other 49 countries/regions, respectively. As a result, the USA plays the most essential part in the field of UFS, as it has the most collaborative publications with international partners, for a total of 1250 publications with the other 49 countries. Furthermore, it has more cooperation with the UK and China than with any other countries. As a country becomes more important on the global stage, it will tend to engage in more academic cooperation with other countries/regions.

### 3.3. Institution Contribution Analysis

Figure 4 shows the top 20 most productive institutions. Among the 5390 institutions involved in UFS research, only 26 (0.46% of total) institutions published more than 30 papers. The USA has the largest share of the selected institutions, with 11, followed by China with four. The Chinese Academy of Sciences published the most papers (139), far beyond that of any other institution, and its volume of publications has increased significantly in recent years. This is followed by the University of Sao Paulo (90), the University of Minnesota (52), Beijing Normal University (45), and the University of Illinois (43). The h-index results show that the Chinese Academy of Sciences (27), the University of Minnesota (22), Harvard University (20), and the Universities of Ghent, of Sao Paulo and of Illinois (all 19) are the most influential academic institutions. Furthermore, the 100 institutions with the largest volume of publications were selected for collaborative analysis (Figure 5). Among these institutions, the University of Oxford, University of Minnesota, University of Copenhagen, Chinese Academy of Sciences, and University of Sao Paulo are the most collaborative. Each of these institutions has cooperated with 35, 31, 28, 26, and 25 of the other 99 institutions, respectively.

### 3.4. Research Shifts and Journal Activity

According to WoS’s classification for subjects, there is a total of 252 subject categories. UFS research has been associated with about 187 subject categories between 1991 and 2020, indicating that UFS includes a wide range of research areas. In the initial period, there were 89 subject categories related to UFS, with nutrition dietetics (64 publications), environmental sciences (47), and food science technology (42) being the categories with the largest numbers of publications. During the second period, the number of research categories increased to 138, and nutrition dietetics (191) still ranked first. Multidisciplinary agriculture and water resources, which did not appear in rankings in the first period, were added to the top 10. The rankings of public environmental occupational health, environmental engineering, and ecology also rose. In the third period, environmental sciences (1356) ranked the highest and has occupied a large proportion, 32.6%. Green sustainable science technology, which even did not appear in the first and second period lists, ranked second (598). In addition, environmental studies and energy fuels were added to the rankings. Figure 6 shows the changes in the structure of the research categories. It can be seen that research on environmental fields is getting more attention. Researchers are also concerned about diet and nutrition. Whereas, the research related to agriculture were relatively reduced, such as agronomy and agriculture multidisciplinary.

The selected 5360 publications were published in 1594 different journals, but most of the journals (63.6%) only published one UFS-related paper. Table 2 shows the 20 most productive journals during 1991–2020, covering 27.5% of the total publications related to UFS. Sustainability is the most productive journal, with 202 publications, followed by the Journal of Cleaner Production (148), and Public Health Nutrition (140). Among the selected journals, Resources Conservation and Recycling has the highest IF value, 8.086, with 40 publications. In addition, we conducted a citation analysis of these journals. LCS refers to the number of times the publications have been cited in the current database, and we ranked the journals according to the LCS value. The results show that the top six cited journals are among the top 20 productive journals, and Journal of Cleaner Production ranked first, indicating that it has certain influence in the field of UFS. However, some journals, such as PLOS One and BMC Public Health, were not cited by any of the current database publications, despite their high volume of publications.

These journals have different research priorities. For example, Sustainability is an international cross-disciplinary journal of environmental, cultural, economic, and human social sustainability; the Journal of Cleaner Production is concerned mainly with cleaner production, the environment, and sustainability research and practice; Public Health Nutrition focuses on issues related to nutrition. In addition to the fact that UFS study has attracted more attention from researchers in various fields, the general increase of new scientific journals is also a major reason why there is more research on UFS. There were only a few journals that published UFS-related publications in the 1990s, such as Food Policy. Though the distribution shows excellent diversity in journals, the most productive journals are similar to the discipline categories, closely related to environment and nutrition.

### 3.5. Keywords and Highly Cited Publications Analysis

There are 11,776 keywords in the selected publications. In total, 35 keywords appeared only once during the first period; hence we did not list these (Table 3). The number of keywords has increased significantly over time, with the third period having about 7441 more keywords than the second period. For instance, only 17 papers covered ‘urban agriculture’ in the second period; occurrences of this phrase increased to 208 in the third period, ranking second. The third-ranking keyword in the third period was food waste (146), which did not appear during the second period. Sustainability, notably, has always appeared on the list of high-frequency use. Remarkably, China ranked in the top five in the frequency of keyword occurrences during the second and third periods. As for changes in the frequency of keywords in the second and third periods, food security (37 occurrences in the stable-growth period, and 201 occurrences in the rapid-growth period) is the most crucial topic in UFS research—not surprisingly, as food is one of the most basic human needs [36]. The rapid development of urbanization has further aggravated the shortage of agricultural land; therefore, ensuring food security in the face of rapid urbanization is an urgent problem that must be explored and solved [14,37]. The second period was more inclined to consumer-related topics, such as food consumption, diet, and obesity issues, while in the third period, there was more research on food production and food waste. Population growth and the reduction of arable land have prompted research on producing food more efficiently [38,39,40], and food waste permeates the entire food supply chain and has adverse effects on resources, the environment, and society [41].

We analyzed keywords that appeared at least 10 times over the whole research period. A total of 266 keywords met this requirement, and they can be roughly divided into three clusters. As shown in Figure 7, each circle represents a keyword; the size of the circle represents the frequency of the keyword, and the line between the circles indicates that they appear together in a publication. Thus, based on the clustering and co-occurrence analysis of high-frequency keywords, we can clearly identify popular and critical research areas. The three clusters are mainly focused on the different subsystems of UFS.

In Cluster I, research related to food production has been primary; the keywords ‘food security’ (238 occurrences), ‘urban agriculture’ (208), and ‘sustainability’ (137) are highlighted. Food security is defined as the condition in which all people, at all times, have physical and economic access to sufficient safe and nutritious food to meet their dietary needs and food preferences for a healthy and active life [42]. Whether in developed or developing countries, food security is predominantly an urban issue related to a city’s sustainability (132) [43,44]. It requires action on multiple aspects, such as efficient production [22], healthy eating habits [45], and well-planned governance [46,47,48]. In the production process of ensuring adequate food supplies, urban agriculture is an inevitable trend for cities’ future development (195) [17,49,50]. Moreover, urban agriculture could mitigate climate change (78) by reducing carbon emissions from food transportation [17]. Keywords in Cluster II are mainly related to food consumption (129), such as diet (129) and nutrition (105). Rising incomes and urbanization have led to a global dietary change characterized by a high intake of meat and refined sugars, and less fiber [18,51]. Obesity (127) caused directly by an unhealthful dietary structure is widely studied [51], and researchers have paid particular attention to food consumption among children (82) and adolescents (54) [52]. In addition, food safety (87) is also an issue of concern to many people. Indeed, food safety (84) and food security are interlinked and distinct [53]. In general, food safety refers to the assurance of food quality for consumer safety, and diverse sources of food supply, as well as environmental pollution, will significantly affect food safety. Cluster III is mainly divided into two themes. One is “food waste” (146)—a hot issue since a quarter to a third of the world’s food is wasted [54], creating the need for a tremendous amount of food waste to be disposed of. Anaerobic digestion (78) is a promising technology for food waste treatment compared to traditional treatment methods, such as composting (18), and incineration (15) [55,56]. The other theme mainly entails the support of tools and methods, such as life cycle assessment (63), input–output analysis (12) and material and substance flow analysis (25). These methods are commonly used and mature in the field of industrial ecology (39) [57,58,59].

Table 4 lists the top-cited publications in the field of UFS in terms of LCS. In this study, a high LCS of an article indicates that it is essential literature in the field of UFS. Compared with GCS (the number of times this article was cited in the whole WoS database), LCS is more valuable for reference in research in specific fields. As is shown in Table 4, the most highly cited article is entitled “Global nutrition transition and the pandemic of obesity in developing countries”, published in Nutrition Reviews in 2012, with an LCS of 79 and a GCS of 1742. This review article documented the links between diet and obesity [60]. It analyzed obesity in the context of income differences and urban–rural differences and pointed out that the prime focus must be on the food supply and on improving dietary quality to prevent obesity. The second and third highest ranked articles were all related to urban agriculture.

## 4. Discussion

### 4.1. Research Hotspots and Trends

The UFS research involves approximately 74% of all subject categories, which means there are numerous research perspectives on UFS. Although the UFS covers a wide range of subject categories, a large proportion of the publications were related to environment and nutrition. This is because environmental and nutrition issues are not only closely related to the food system, but also hot issues that are currently attracting attention. The multiple environment impacts that challenge the stability of the Earth system are primarily caused by food production and consumption [68,69]. Additionally, nutrition is a global challenge; nearly every country faces public health challenges, whether malnutrition or overweight [70]. Thus, it has led to an increase in the number of journals in these fields, such as Sustainability, Journal of Cleaner Production, and Public Health Nutrition. Nevertheless, as the results show, a large number of publications does not necessarily mean that it has a strong influence; it may appear due to the large volume of journal publications. The main reason for the declining trend of agriculture-related disciplines in UFS is that our research scale is limited to cities, which are not the main development areas of agriculture, and it mainly focuses on the production side. While this does not mean that urban agriculture should be ignored. What is more, the UFS-related research has gone beyond the scope of a single subject. There is a phenomenon of mutual integration and connection between disciplines. Interdisciplinary is a research mode centered on problem-solving, which promotes the solution of many important practical problems [71]. Additionally, since the urban food system involves extensive research, interdisciplinary research is necessary. Therefore, interdisciplinary research, waving environment, nutrition, ecology, and so on, need to be considered to form a better research mechanism.

Results based on keyword analysis provide clues for future development and can identify gaps in UFS research. Since the UN specified the SDGs in 2015, sustainability has drawn wide concern. In the context of economic transformation and climate change, there have been many UFS studies related to sustainability [72,73]. To promote the sustainable development of UFS, it is necessary to define how to measure and assess the sustainability of urban food systems. To date, the sustainability assessments of UFS have not been well understood. Although some studies have established the sustainability assessment framework of UFS, most of these remain in the theoretical stage and are difficult to apply in practice. Therefore, it is necessary to establish a set of reasonable assessment indicators to assess the sustainability of UFS comprehensively.

Furthermore, the resilience of UFS is significant but less frequently considered. In fact, resilience is complementary to sustainability [74]: sustainability focuses on perpetuating system functions, while resilience is the system’s response to unforeseen disturbance. In 2020, the COVID-19 pandemic spread across the globe, posing a severe threat to food security, mainly reflected in food shortages and delayed supply caused by impeded logistics. It has been a wake-up for thinking about the resilience of supply chains and future food systems. [75,76,77]. The study suggested that the COVID-19 pandemic may add an additional 83–132 million people to the undernourished ranks in 2020 [2]. Urban agriculture is an effective way to meet urban food supply needs and ensure urban food security under such emergencies. Although urban agriculture has received widespread attention, compared with traditional agriculture—which already has good mechanisms after generations of development—there are still many unsolved urban agriculture problems, such as policy formulation and management gaps. Research has shown that urban agriculture’s geographical specificity requires that policy formulation be based on urban agriculture typologies that vary across forms and functions [78]. Diversity is considered to be the key to making UFS more resilient. When developing urban food production, it is necessary to seek out various food sources and avoid over-dependence on any single source. A previous study has suggested that sometimes the localization of food production may increase the food supply chain’s overall environmental impact [79]. Therefore, there may be many trade-offs between sustainability and resilience, and how to balance the sustainability and resilience of UFS is an issue worth exploring.

Each subsystem of the urban food system needs an innovative transformation to adapt to the current global situation. Innovation includes theoretical, institutional and technological innovation, which complement and influence each other. Technological innovation inevitably involves trade-offs among many desirable objectives, and its orderly development needs to be coordinated through regulations and sociocultural norms [80]. Studies have shown that the speed of innovation could be significantly increased by appropriate incentives, regulation and social license [81]. The outbreak of the COVID-19 pandemic has led to a boom in innovation, especially digital innovation. E-commerce online platforms and delivery services have created favorable conditions for food security in the city. In the processing and transportation stage, emerging fields such as cold chain [82], intelligent packaging [83], and smart logistics [84] also have higher technological innovation requirements. With the accelerated pace of life, people continue to pursue more convenient and rapid methods of food access and consumption. As a result, the takeout industry is emerging as a new urban economy, although food safety issues and environmental pollution issues such as takeout containers need to be managed and controlled by innovative systems and technologies. Therefore, there is an urgent need to recognize the obstacles and opportunities for comprehensive urban food system changes. The sustainable and resilient development of UFS should be promoted through innovative research.

### 4.2. Challenges of UFS in Developing Countries

The study found that UFS research mainly occurs in developed countries, such as the USA and the UK (accounted for 34.5%), which have developed economies and invested heavily in scientific research. However, a few studies in developing countries may limit the generalizability of findings for UFS research and belie a multiculturally and globally relevant viewpoint. While, China is an exception and has put much emphasis on UFS research. Indeed, compared with developed countries, urban population growth in developing countries may be more significant due to rural–urban migration, which has put urban food systems under enormous pressure [85]. The urban poor in developing countries are particularly vulnerable to food insecurity [16]; unlike the rural poor, they have less access to land for farming, making it harder for them to produce their own food when food is in short supply or they cannot afford to buy it. Additionally, while the urban poor face significant challenges in accessing any kind of food, challenges are even more formidable in terms of obtaining healthful food [86]. In addition, the co-occurrence analysis indicated that cooperation among developed countries is frequent. There is relatively little cooperation between developing and developed countries except China, implying that it is necessary to strengthen research in developing countries, and more cooperation should be encouraged between developing and developed countries, to find common ground and solve problems, and more mature methods or models could be applied to developing countries through global cooperation.

Limited data availability often makes it difficult to conduct research in some countries, particularly low-income countries. Some countries’ databases are incomplete, so the accurate and reliable data cannot be found in many places, restricting research. Additionally, the previous had similar findings [87]. Thus, more reliable database management tools need to be developed. Moreover, international organizations, such as the United Nations, could reasonably promote more complete databases and update information sharing. For example, some primary data on local food production, food trade or food consumption may be collected only sporadically and quickly become obsolete; these need to be updated and maintained regularly. In addition, at present, most studies on UFS are focused on a single city, and there is a lack of comparative studies among cities, especially among cities in countries with different levels of development. The stages of urban food systems in developing and developed countries differ in terms of current problems. For example, urban food waste in developing countries is mainly due to imperfect infrastructure and aging transportation technologies that cause a large amount of perishable food to be wasted before being consumed, while food waste in developed countries occurs more often in the consumption stage [88]. By studying the UFS in different development stages, we can find the commonalities and characteristics, and then more accurately summarize the regularity, and put forward reasonable improvement measures and policy suggestions.

### 4.3. Limitations

Although this study objectively reveals the current status and trends of UFS research, several limitations should be discussed. The first limitation arises from our reliance on WoS as the source of documents, and did not include other databases such as Scopus, and Medline. Although WoS is the most widely adopted database in visualization research [89,90], the scope of data collection may be limited. Another limitation is related to new papers and works that are rarely cited. New research requires a certain amount of time to accumulate citations, and the earlier publications tend to have more citations. In addition, since UFS research covers a wide range of fields, it is difficult to conduct a detailed overview of a topic, which will be the focus of our future work.

## 5. Conclusions

The number of studies and articles on UFS has multiplied in recent years, mainly because of the acceleration of global urbanization and the vulnerability of the food system to social and environmental changes. This review is the first of its kind to systematically review the status and trends of UFS research by using bibliometric methods based on 5360 publications obtained from the WoS core collection.

A significant increase and changes in UFS-related interdisciplinary categories were quantitatively revealed, with environment-related becoming the mainstream. Through the co-occurrence analysis of research countries and institutions, we found that most research is performed in developed countries and that these countries have cooperated closely with each other, with China being the only exception. The Chinese Academy of Sciences is the institution with the largest volume of publications. More importantly, the keywords analysis, including co-occurrence analysis and cluster analysis, revealed the current research focus and trends. Researchers tend to focus on three aspects: food security related to food production; food consumption and dietary health; food waste and waste disposal research.

From the systematic analysis of published research of the past three decades, some insights and future research directions about UFS have been identified. First, the current UFS research urgently needs to broaden its reach and establish a comprehensive database to strengthen research in developing countries. Second, there are few studies on UFS resilience and a lack of comparative studies on the trade-offs between resilience and sustainability. Additionally, UFS innovation is an area that needs more focus in the future.

## Figures and Tables

**Figure 1 foods-10-00662-f001:**
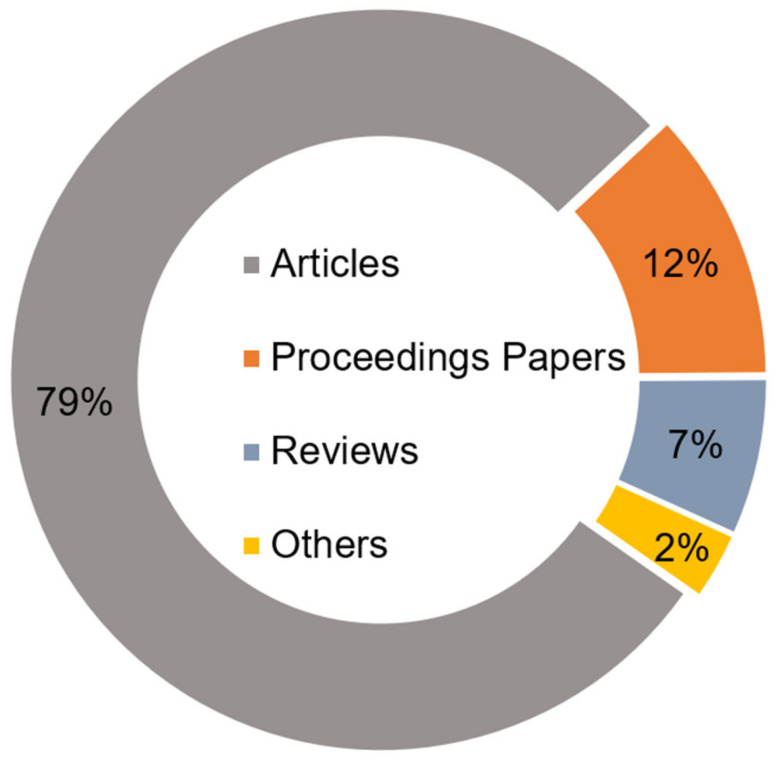
Proportions of each type of Urban food system (UFS) research.

**Figure 2 foods-10-00662-f002:**
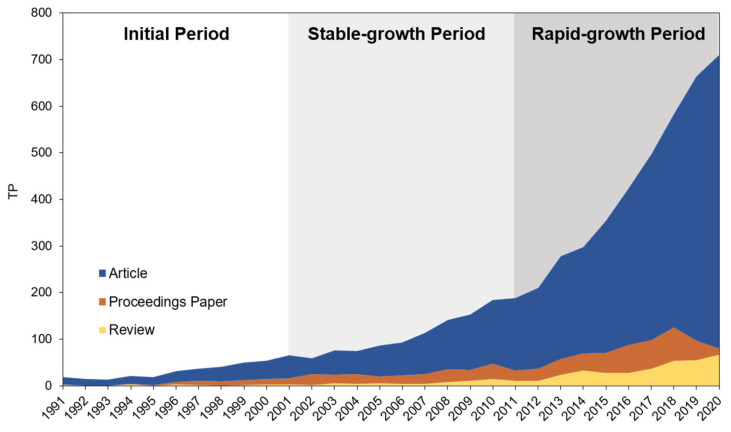
Publishing trend in the area of UFS.

**Figure 3 foods-10-00662-f003:**
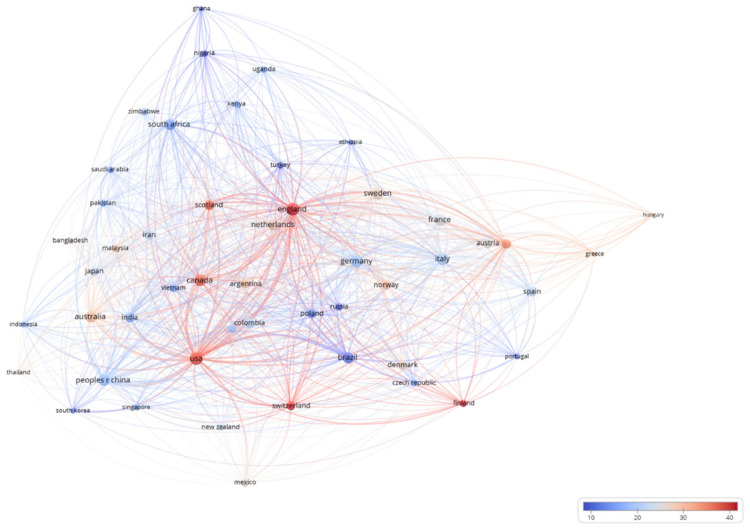
Co-occurrences of countries/regions.

**Figure 4 foods-10-00662-f004:**
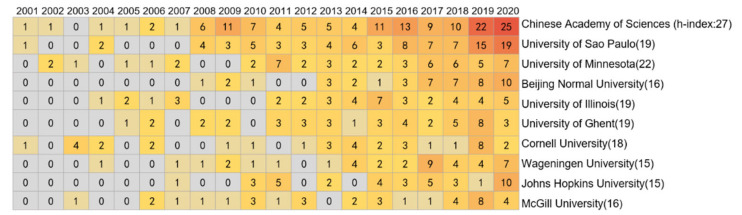
Performance of the top 10 most productive institutions.

**Figure 5 foods-10-00662-f005:**
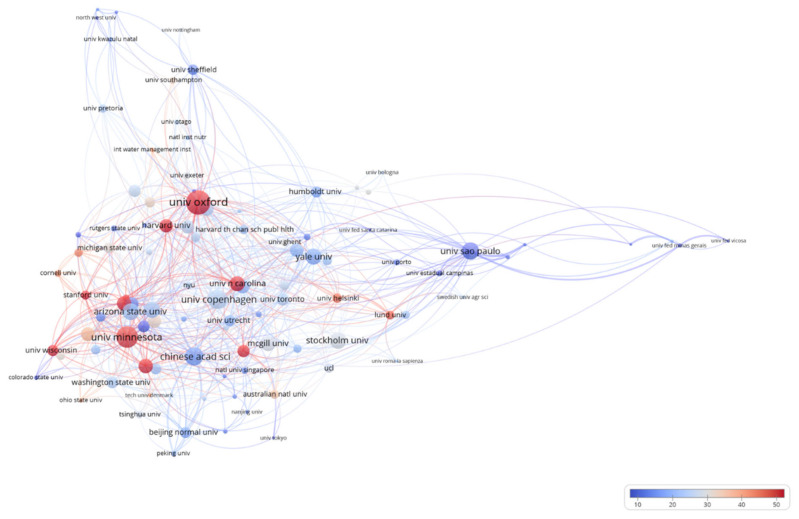
Co-occurrences of the institutions.

**Figure 6 foods-10-00662-f006:**
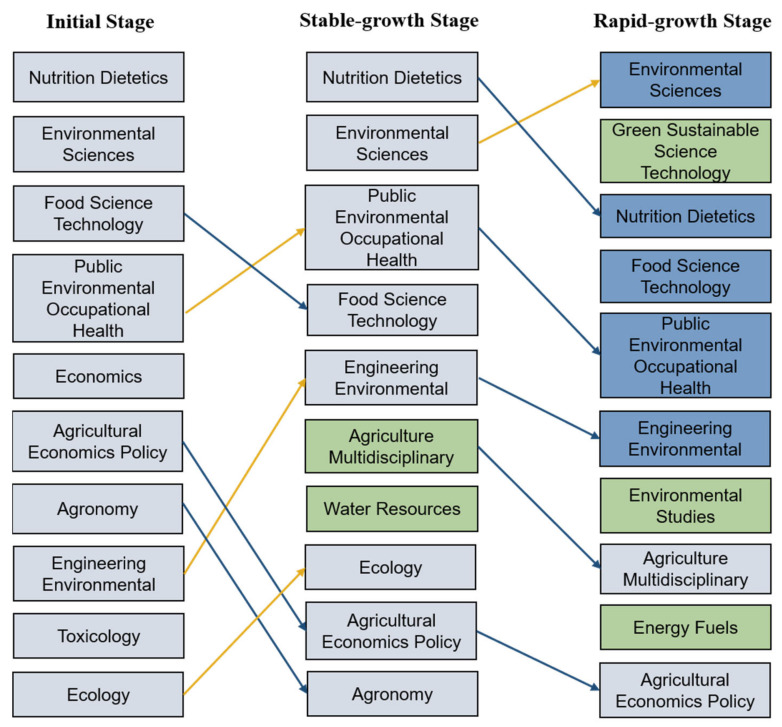
Evolution of WoS categories of UFS research over three periods.

**Figure 7 foods-10-00662-f007:**
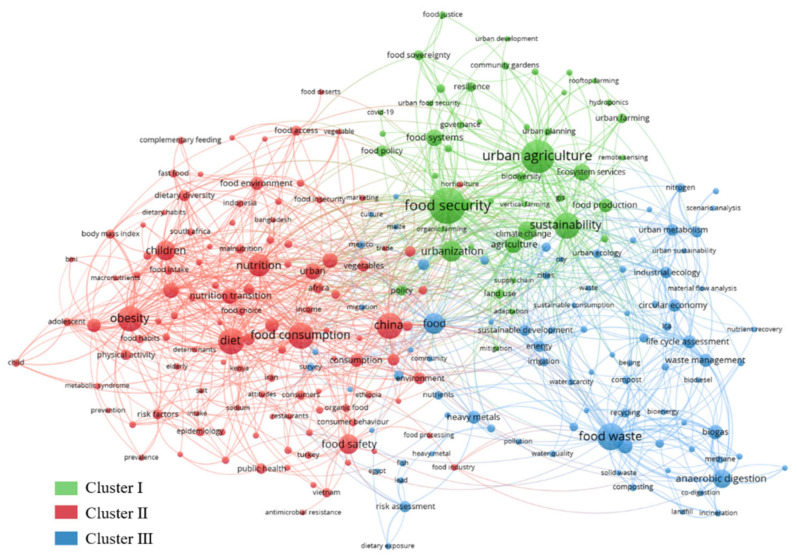
Authors’ keyword cluster analysis.

**Table 1 foods-10-00662-t001:** The top 10 most productive countries/regions over the three periods.

Initial Period				Stable-Growth Period				Rapid-Growth Period			
Country	TP	UR	RDE	Country	TP	UR	RDE	Country	TP	UR	RDE
USA	92	77.42	2.52	USA	247	80.01	2.63	USA	1148	81.68	2.75
Canada	25	77.93	1.71	China *	85	43.17	1.32	China *	619	55.47	2.03
India *	14	26.72	0.70	UK	73	80.04	1.61	UK	372	82.62	1.67
UK	11	78.38	1.59	Canada	57	80.27	1.94	Brazil *	323	85.75	1.23
Finland	11	81.06	2.83	Brazil *	51	82.97	1.05	Canada	269	85.71	2.05
France	10	75.01	2.13	Australia	50	84.64	2.11	Australia	252	69.57	1.33
Italy	10	66.95	0.99	France	41	77.25	2.11	Italy	244	81.27	1.71
Brazil *	9	77.96	1.05	Germany	38	76.07	2.51	Germany	229	77.23	2.91
Japan	9	78.07	2.83	India *	38	29.41	0.79	Netherlands	190	32.82	0.70
Sweden	8	83.74	3.37	Netherlands	38	82.84	1.73	Spain	183	79.61	1.25

TP: total publications; UR: urbanization rate (%); RDE: research and development expenditures (%); * developing countries.

**Table 2 foods-10-00662-t002:** Top 20 productive journals during 1991–2020.

Journal	TP	Percentage (%)	IF	LCS	Rank
Sustainability	202	3.77	2.576	18	63
Journal of Cleaner Production	148	2.76	7.246	248	1
Public Health Nutrition	140	2.61	3.182	160	6
Science of The Total Environment	101	1.88	6.551	191	4
PLOS One	87	1.62	2.740	0	N/A
International Journal of Environmental Research and Public Health	85	1.59	2.849	1	252
British Food Journal	73	1.36	2.102	65	19
Waste Management	69	1.29	5.448	196	3
Appetite	58	1.08	3.608	65	18
BMC Public Health	57	1.06	2.521	0	N/A
Nutrients	54	1.01	4.546	12	86
Agriculture and Human Values	52	0.97	2.442	218	2
Food Security	51	0.95	2.095	74	15
Food Control	47	0.88	4.258	33	37
Food Policy	47	0.88	4.189	170	5
Environmental Science and Pollution Research	43	0.80	3.056	21	53
Asia Pacific Journal of Clinical Nutrition	41	0.77	1.236	36	32
Resources Conservation and Recycling	40	0.75	8.086	83	13
Journal of Environmental Management	39	0.73	5.647	67	17
Urban Forestry Urban Greening	39	0.73	4.021	112	11
European Journal of Clinical Nutrition	36	0.67	3.291	18	63

Rank: Ranking according to the number of citations in the current database.

**Table 3 foods-10-00662-t003:** The 20 most frequently used keywords over the two periods.

Stable-Growth Period	TP	Rapid-Growth Period	TP
Food security	37	Food security	201
Diet	29	Urban agriculture	191
Food consumption	25	Food waste	137
China	23	Sustainability	117
Food	22	China	114
Children	22	Obesity	106
Obesity	21	Food consumption	104
Sustainability	20	Diet	99
Urban agriculture	17	Nutrition	96
Urban	15	Food safety	75
Urbanization	15	Urbanization	75
Irrigation	15	Anaerobic digestion	74
Dietary intake	15	Food	71
Consumption	14	Climate change	70
Adolescents	14	Ecosystem services	67
Food production	13	Children	60
Poverty	12	Food systems	59
Nutrition transition	12	Agriculture	56
Food safety	12	Municipal solid waste	48
Environment	11	Nutrition transition	46

**Table 4 foods-10-00662-t004:** Top 20 local cited references from 1991–2020.

TI	SO	DT	PY	LCS	GCS
Global nutrition transition and the pandemic of obesity in developing countries	Nutrition Reviews [60]	Review	2012	79	1742
Strawberry fields forever? Urban agriculture in developed countries: a review	Agronomy for Sustainable Development [61]	Review	2014	57	161
Urban agriculture of the future: an overview of sustainability aspects of food production in and on buildings	Agriculture and Human Values [17]	Article	2014	52	153
Global diets link environmental sustainability and human health	Nature [18]	Article	2014	47	1003
Food consumption trends and drivers	Philosophical Transactions of The Royal Society B-Biological Sciences [62]	Review	2010	46	753
Exploring the production capacity of rooftop gardens (RTGs) in urban agriculture: the potential impact on food and nutrition security, biodiversity and other ecosystem services in the city of Bologna	Food Security [63]	Article	2014	44	96
Westernization of Asian diets and the transformation of food systems: Implications for research and policy	Food Policy [64]	Article	2007	42	361
Urbanization and its implications for food and farming	Philosophical Transactions of The Royal Society B-Biological Sciences [65]	Review	2010	42	263
Global consequences of land use	Science [66]	Review	2005	40	5934
Reducing greenhouse gas emissions with urban agriculture: A Life Cycle Assessment perspective	Landscape and Urban Planning [67]	Article	2013	40	93

TI: title; SO: source; DT: document type; PY: published year; LCS: local citation score; GCS: global citation score.

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
