# Peer review of "Urban Food Systems: A Bibliometric Review from 1991 to 2020"

_foods, 2021, doi:10.3390/foods10030662_

Round 1

Reviewer 1 Report

An interesting work. The article has been written clearly. It is properly structured. The results and discussion sections are clear and relevant. The conclusions are supported by the results. The bibliography includes study-relevant and up-to-date references. 

Author Response

Thank you for the review and comments. Please see the attachment.

Reviewer 2 Report

This paper is well-organized, well-written, and well-researched. It is very informative and a good addition to this journal; however, the end of the paper falls a bit flat in my estimation.

My main recommendation is that the authors push the results a bit further at the end of the paper to highlight the areas of current research, future research, and gaps in their analysis. While they presented a clear and compelling case to showcase research conducted in many journals, they had to make certain decisions in terms of how to conduct the analysis (limiting the scope of the number of journals represented as well as an overemphasis on certain types of journals that have more rapid or public health emphasis). Does this necessarily mean that they are most influential journals, or do they speak to broader shifts in the field, or are their structural reasons why certain types of journals have higher citation counts? In order words, a more numerical citations analysis can have value but there is also value in exploring the various disciplinary trajectories to explain why certain subfields or areas of interest become more highly cited or highly valued at certain points. These comments are all to say that the author can be more critical of their own analysis at the end in order to explain the value but also limitations of a literature review study developed in this way.

To this point, I think the discussion section is not sufficient in its current form as it needs to more critically examine the findings in light of my comments above. Right now, the discussion is a bit too superficial, as it does not examine the real meaning and limitations beyond the findings and current/future research. The discussion section should not be guessing at reasons why the results are but rather base the analysis in a more in-depth discussion of the works. The authors have not critically analyzed the lit review findings within the context of their in-depth knowledge of the field.

Additionally, the conclusion section seems somewhat misplaced or not clear in terms of its value to the manuscript.

These improvements are important and would significantly enhance the manuscript.

Lastly, the authors should read through the manuscript in order to maximize readability and language clarity.

Author Response

Thank you for the review and comments. The constructive comments really help to improve the research and polish the manuscript. Please see the attachment.

Reviewer 3 Report

The manuscript is a good summary of the current situation in the scientific area. It is also important the authors noted the lack of specific research. I consider the recommendations of the authors that the scientific community should focus more on the study of sustainability x resilience in the future as very relevant.

The authors use the same words in their keywords as in the title of the article. If they are interested in better searchability, I recommend using other words in keywords.

Author Response

(The authors gave the same response as above.)

Round 2

Reviewer 2 Report

The authors have responded to the reviewers' comments. My only two suggestions at this point are to re-read the end of the paper to ensure that its impact and overall assessment are as clear and strong as possible. Also, although well-written, the paper could be re-read as a whole to maximize clarity and overall flow.

Good job.

Author Response

Thanks again for the review and suggestions. Please see the attachment.
